Plastic debris in plastic-mulched soil—a screening study from western Germany

Steinmetz Zacharias steinmetz-z@uni-landau.de
Schröder Heike
iES Landau, Institute for Environmental Sciences, Group of Environmental and Soil Chemistry, University of Koblenz–Landau , Landau , Germany
Mortimer Monika
Electronic publication date: 2022 Jul 19
Publication date: 2022
Volume: 10
Electronic Location ID: e13781
Received 2022 Apr 1; Accepted 2022 Jul 3
Copyright: ©2022 Steinmetz and Schröder
Copyright year: 2022
Copyright holder: Steinmetz and Schröder
License: This is an open access article distributed under the terms of the Creative Commons Attribution License, which permits unrestricted use, distribution, reproduction and adaptation in any medium and for any purpose provided that it is properly attributed. For attribution, the original author(s), title, publication source (PeerJ) and either DOI or URL of the article must be cited.
License URL: https://creativecommons.org/licenses/by/4.0/

Keywords: Plastic mulching, Agriculture, Microplastics, Py–GC/MS, FTIR

Funding: University of Koblenz–Landau (project “PLAST”) This research was financially supported by the pilot program “Profil3” of the “Bildung Mensch Umwelt” research fund at the University of Koblenz–Landau (project “PLAST”). The funders had no role in study design, data collection and analysis, decision to publish, or preparation of the manuscript.

==============================
Background

Agricultural plastic mulches offer great benefits such as higher yields and lower pesticide use. Yet, plastic mulches may disintegrate over time and fragment into smaller debris. Such plastic debris is expected to remain in the field after removal of the plastic mulch and thus contributes to soil contamination with plastics.

Method

To investigate this, we collected soil samples at 0–10 cm and 10–40 cm depth from three fields covered with black mulch film for three consecutive years. Three fields without any reported plastic use served as control. Visual plastic debris > 1 cm (macroplastics) was collected from the soil surface. Mesoplastics between 2 mm and 1 cm were density separated from the sampled soil using saturated NaCl solution and analyzed by Fourier-transform infrared spectroscopy. Debris ≤ 2 mm (microplastics) was dispersed from 50 g soil using sodium hexametaphosphate solution followed by the aforementioned density separation. The separated polyethylene (PE), polypropylene (PP), and polystyrene (PS) were quantified via solvent-based pyrolysis–gas chromatography/mass spectrometry (Py–GC/MS).

Results

With 89–206 fragments ha−1, the majority of macroplastics were located in fields previously covered with mulch films. 80% of the collected specimen were identified as black PE film. The number of mesoplastics in plastic-mulched soil was 2.3 particles kg−1, while only 1.0 particles kg−1 were found in the reference fields. Py–GC/MS revealed microplastic levels of up to 13 mg kg−1. The PE content was significantly higher in plastic-mulched fields than in reference fields.

Discussion

Although the identified plastic levels are lower than those reported in comparable studies, our results still suggest that plastic mulching functions as a source of plastic debris in agricultural systems. Due to its severely restricted degradability, these plastics are likely to accumulate in soil in the long term and further fragment into smaller and smaller debris.

Introduction

Ensuring food security in the course of mounting global crises has become a major challenge of our age (Godfray et al., 2010). To feed an ever growing world population, agricultural plastic mulching is considered an effective measure for increasing yields while reducing water and pesticide use (Steinmetz et al., 2016). These prospects have led to about 460,000 ha of European cropland covered with plastics in recent years (Razza, Guerrini & Impallari, 2019). However, plastic mulches are mechanically stressed and embrittle while in use (Astner et al., 2019), which makes it difficult to completely remove them from the field at their end of life. Thereby, parts of the applied mulch films are likely to remain in the field where they eventually fragment into smaller debris, accumulate, and potentially impede soil functioning (Steinmetz et al., 2016; Hurley & Nizzetto, 2018; Rillig, Leifheit & Lehmann, 2021a).

Yet, the extent to which agricultural plastic mulches emit plastic debris into their surrounding is incompletely understood. So far, the majority of studies screening for plastic debris in plastic-mulched soil have been conducted in China and attracted attention with reporting particularly high plastic levels of 1,000–42,000 particles kg−1 (Huang et al., 2020; Zhang & Liu, 2018). Such extreme values are probably associated with decades of continuous plastic mulching (Huang et al., 2020; Liu et al., 2018). In Europe, comparable screenings are sparser and the reported particle counts are as low as 4 ± 11 plastic particles kg−1 (Harms et al., 2021). Crop rotations and shorter plastic applications of several months to about three years appear to limit plastic emissions. Moreover, Steinmetz et al. (2022) recently associated elevated plastic contents in the vicinity of previously mulched soil with the use thinner (40 µm) and perforated films while ≥ 50 µm thick mulches were not indicated to emit any plastics. However, the aforementioned screening studies often lacked a proper non-mulched control, excluded particles < 1 mm, or restricted their sampling to the uppermost 5–10 cm of the soil. Thus, more comprehensive and comparative studies are required, particularly in Europe.

To address this research gap, our study aimed at systematically assessing macroplastics (> 1 cm), mesoplastics (1 cm to 2 mm), and microplastics (≤ 2 mm) in the topsoil (0–10 cm) and subsoil (10–40 cm) of three fields previously covered with plastic mulch as compared with non-mulched controls in western Germany. Notwithstanding common size classifications of plastic debris (Hartmann et al., 2019), we chose 2 mm instead of 1 mm as the upper size limit for microplastics to be consistent with soil analyses usually performed on fine soil ≤ 2 mm (Thomas et al., 2020). After density separation with saturated NaCl solution, plastic debris > 2 mm was analyzed via Fourier-transform infrared (FTIR) spectroscopy with attenuated total reflection (ATR). Polyethylene (PE), polypropylene (PP), and polystyrene (PS) microplastics were quantified by solvent-based pyrolysis–gas chromatography/mass spectrometry (Py–GC/MS). We hypothesized that plastic mulches function as a source of plastic debris and thus lead to higher plastic levels in plastic-mulched soil than in non-mulched controls. Furthermore, we assumed that plastic fragments are translocated to deeper soil layers through plowing, percolation, or bioturbation (Huerta Lwanga et al., 2016; Rillig, Ziersch & Hempel, 2017).

Methods

Study area and experimental design

The field study was conducted west of the Rhine River in the Lower Rhine Bay between Köln and Düsseldorf (North Rhine-Westphalia, Germany, Fig. 1). The local climate is temperate with a mean annual temperature of 10.5 °C and a total annual precipitation of 770–920 mm (LANUV NRW, 2019).

Figure 1 Sampling scheme in the study area (satellite imagery ©2022 TerraMetrics/Google); on the map of Germany, the study area is highlighted in red.

We studied six conventionally managed fields of 0.1–8 ha. Sites 1–3 were located near Sinnersdorf (51°1′N 6°47′E), Nettesheim (51° 3′N 6°42′E), and Kaarst (51°12′N 6°36′E). Sites 1 and 2 were cultivated with strawberries (Fragaria x ananassa) continuously covered with black PE mulch (50 µm film thickness) for the previous three years. Site 3 was planted with zucchinis (Cucurbita pepo) and mulched with black PP of the same film thickness during the last three growing seasons. Sites 4–6 were situated near Pulheim (51°0′N 6°47′E), Stommelerbusch (51°4′N 6°46′E), and Rommerskirchen (51°2′N 6°42′E), and served as reference sites without any previous plastic use. Sites 4 and 6 were cornfields (Triticum aestivum); site 5 was cultivated with potatoes (Solanum tuberosum). None of the six sites has received any compost or sewage sludge applications in recent years. The closest urban or industrial areas were at least 5 km away from the experimental sites. While this explicitly excluded some other potential sources of plastic inputs, littering and wind drift could not be controlled for.

Sampling procedure

In autumn 2020, after harvest and retrieval of the plastic covers, the sites were sampled along four evenly spread diagonals of equal length (Fig. 1). The soil sampling was granted by the respective land owners. Macroplastic fragments > 1 cm were collected from surface soil at a maximum distance of 1 m from either side of the diagonals, based on the four-eye principle to decrease the risk of overlooking plastic debris (Prume, Gorka & Löder, 2021). For the quantification of mesoplastics (2 mm to 1 cm) and microplastics (≤ 2 mm), soil was sampled from five equidistant sampling points along each diagonal transect totaling to 20 soil samples per site. While the topsoil (0–10 cm) was removed with a stainless steel core cutter (5 cm diameter), the subsequent subsoil (10–40 cm) was sampled using a stainless steel drill rod (2 cm diameter). Topsoil and subsoil were each pooled in metal buckets to obtain representative composite samples of > 2 kg.

Soil characterization

The soil samples were dried at 40 °C for 72 h, manually homogenized, and sieved at 2 mm using a stainless steel mesh. The soil texture on site was determined using a soil hydrometer in accordance with ASTM D422-63 (2007). Soil organic carbon (Corg) was quantified by dry combustion elemental analysis (Vario MICRO Cube, Elementar, Germany) as detailed in DIN EN 15936 (2012). The soil pH was measured in the field using a handheld probe (X4 Life 4in1, Braunschweig, Germany).

Extraction of plastic particles from soil

Since agricultural plastic covers are primarily made of light-density polymers like PE and PP, mesoplastics were density-separated from non-sieved soil with a saturated NaCl solution (1.2 g cm−3). Polymers with a higher density than the applied NaCl solution, such as PET or PVC, were thus systematically excluded from our analysis. To this end, 1 kg of topsoil and subsoil were each mixed with 3 L density solution using a stainless steel whisk. After 3 h of sedimentation, the supernatant was passed through a 1 mm stainless steel sieve. Retained suspect particles > 2 mm were picked with stainless steel tweezers and transferred to glass vials for subsequent FTIR–ATR analysis.

Microplastics were extracted in accordance with Steinmetz et al. (2022). In brief, 50 g of sieved soil were dispersed with 125 mL of sodium hexametaphosphate solution (40 g L−1, pH 5.4, CAS 68915-31-1, ≥ 99% purity, Carl Roth, Karlsruhe, Germany) in a 1 L glass separation funnel and agitated at 150 rpm for 2 h to retrieve plastic debris potentially occluded in soil aggregates. The subsequent addition of 90 g NaCl and 125 mL of distilled water produced a saturated NaCl solution with a density of 1.2 g cm−3 which was agitated for another 2 h. After 16 h, the sedimented soil was drained from the funnel, and the remaining supernatant was rinsed through a pleated cellulose filter (Whatman 589/2, 4–12 µm particle retention, GE Healthcare, Buckinghamshire, UK). The filters were transferred to glass culture tubes (GL18, VWR, Darmstadt Germany), dried at 60 °C, and extracted with 8 mL of a 1:1 mixture of p-xylene (CAS 125 106-42-3, > 98.0% purity, Fluka Analytical, München, Germany) and 1,2,4-trichlorobenzene (CAS 120-82-1, 99% purity, Alfa Aesar, Kandel, Germany) at 150 °C for 1 h. Polymer dissolution in p-xylene/1,2,4-trichlorobenzene facilitated sample handling and enabled the representative analysis of sample aliquots. After cooling down, the extracts were transferred to glass vials with polytetrafluoroethylene (PTFE)-sealed caps for subsequent quantification of PE, PP, and PS via Py–GC/MS. The recovery of the extraction procedure was 86–105% (Steinmetz et al., 2022).

Plastic analysis

Visual mesoplastics and macroplastics were photographed, classified by shape, and analyzed via FTIR–ATR for their polymeric composition (Figs. 2A, 2B). FTIR–ATR spectra were acquired from 30 sample scans and eight background scans between 4,000 and 650 cm−1 at a resolution of 4 cm−1. The spectra were identified with Open Specy, version 0.9.3 (Cowger et al., 2021). Library matches exceeding a Pearson’s r of 0.8 were considered reliable and taken into account for further data evaluation.

Figure 2 Flow chart of the sample preparation steps and subsequent analyses of (A) macroplastics, (B) mesoplastics, and (C) microplastics.

Microplastic PE, PP, and PS were quantified using the solvent-based Py–GC/MS method validated by Steinmetz et al. (2022) (Fig. 2C). In brief, 2 µL of sample extracts were pyrolyzed with a Pyroprobe 6150 filament pyrolyzer (CDS Analytical, Oxford, United States) coupled to a Trace GC Ultra with DSQII mass spectrometer (Thermo Fisher Scientific, Bremen, Germany). After purging solvents and volatiles for 3 min at 300 °C, the pyrolyzer was flash heated (10 K ms−1) to 700 °C for 15 s. The formed pyrolysates were chromatographically separated on a 30 m × 0.25 mm capillary column (5% phenylarylene, 95% dimethylpolysiloxane, 0.25 µm film thickness, ZB-5MS, Phenomenex, Aschaffenburg, Germany) at a 1.3 mL min−1 He flow. The GC oven heated from 40 °C to 300 °C at an 8 K min−1 rate. The mass spectrometer selectively monitored m/zs 70 and 126 for the characteristic PP pyrolysate 2,4-dimethyl-1-heptene, m/z 104 for the PS pyrolysate styrene, and m/zs 82 and 95 for the PE n-alkadiene marker 1,21-docosadiene. Deuterated PS was used as internal standard, and its marker compound styrene-d5 was acquired at m/z 109. The chromatograms were evaluated using OpenChrom (Lablicate Edition 1.4.0.202201211106, Wenig & Odermatt, 2010). In line with Steinmetz et al. (2022), calibration curves responded linearly (adj. R2 > 0.989), the measurement repeatability was < 20% relative standard deviation, and the method limit of detection (LOD) was 0.40 mg kg−1. Microplastic contents between plastic-mulched fields and non-mulched controls were normally distributed and homogeneous in variance and thus statistically analyzed using Student’s t tests. Effect sizes were measured with Cohen’s d using R statistical software (version 4.2.0).

Quality control

To reduce the risk of contamination, all sampling and laboratory equipment coming into direct contact with the samples or extract solutions was made of glass, stainless steel, or PTFE. The worn laboratory coats were of 100% cotton. All samples and extracts were kept in closed vessels or covered with aluminum foil or watch glasses whenever possible.

In addition, a potential plastic contamination during sample handling was monitored with procedural blanks, which went through all sample preparation steps but without soil addition.

Results

Soil properties

The soil texture in the investigated fields ranged from silty clay to sandy loam (Table 1). Clay contents leveled at 25–32%. The lowest silt content (22%) and the highest sand content (53%) were found at site 1. By contrast, site 4 showed the highest silt content (65%) and lowest sand content (5%). The soil Corg content was 0.8–1.7%. The topsoil at site 1 and 2 featured the highest Corg contents with 1.6 and 1.7%, respectively. The lowest Corg contents (0.8%) were found in the subsoil of sites 3 and 5. The soil pH was circumneutral ranging from 6 to 7 across all sites.

Table 1 Soil texture and Corg contents at the study sites.

Site	Texture	Clay (%)	Silt (%)	Sand (%)	Depth (%)	Corg (%)	pH	
1	Sandy loam	25	22	53	0–10	1.6	7.0	
					10–40	0.9	
2	Silty loam	27	63	10	0–10	1.7	6.0	
					10–40	1.0	
3	Silty loam	30	56	14	0–10	0.9	6.0	
					10–40	0.8	
4	Silty loam	30	65	5	0–10	1.2	6.0	
					10–40	0.9	
5	Clayey sandy loam	31	25	44	0–10	0.9	6.0	
					10–40	0.8	
6	Silty clay	32	58	10	0–10	1.0	6.5	
					10–40	1.0	

Macroplastics and mesoplastics

A total of 35 macroplastic items were collected from surface soil while following the predefined sampling transects. With respect to the sampled area, this was equivalent to 89–206 fragments ha−1 at sites 1–3, which were previously covered with mulch films. 75 fragments ha−1 were found at the control site 5. Sites 4 and 6 did not show any macroplastic contamination. 80% of the collected items were identified as black PE foils. In addition to that, we found rope fragments, clips, and residues of blue and white plastic films.

The total number of mesoplastic particles extracted from non-sieved soil was 17. Plastic-mulched topsoil and subsoil each contained 2.3 plastic particles kg−1 (Fig. 3). With 1.3 particles kg−1, the majority of particles were films, followed by fragments (0.7 particles kg−1) and fibers (0.3 particles kg−1) of blue and black color (items 1–3, 6, 7, 12–14, 15, and 17, Fig. 4). The predominant polymers were identified as PP, PE, and PS (r > 0.83, Fig. 3). In non-mulched controls, only 1.0 particles kg−1 were found in the topsoil. These were a resin-like bead, a black PE film, and a PE fiber (r > 0.83, Fig. 3; items 8–9, Fig. 4). The non-mulched subsoil was free of mesoplastics.

Figure 3 Mesoplastic counts (2 mm to 1 cm) in mulched soil and non-mulched controls.

Figure 4 Photographs of mesoplastics between 2 mm and 1 cm size retrieved from (A) 0–10 cm topsoil and (B) 10–40 cm subsoil.

Microplastics

Microplastic PE levels ranged from 0.03 to 0.55 mg kg−1 across all sites and thus hardly exceeded the method LOD of 0.40 mg kg−1 (Fig. 5). Yet, mean PE levels of mulched topsoils (0.43 mg kg−1) were significantly higher than the control treatment (Student’s t = −3.156, p = 0.034, df = 4, Cohen’s d = −2.6). PE levels in the subsoil were exclusively below method LOD, irrespective of the treatment (Student’s t = 0.167, p = 0.876, df = 4, Cohen’s d = 0.1). Similarly, PP was only quantified in mulched soil, and averaged 0.29 mg kg−1 in the topsoil and 0.50 mg kg−1 in the subsoil. The respective maximum values were 0.8 and 1.2 mg kg−1. However, due to the high variability of PP levels across sites, the differences between treatments were not statistically significant, neither in the topsoil nor in the subsoil (Student’s t = −0.974 and −1.433, p = 0.307 and 0.225, df = 4, Cohen’s d = −1.0 and −1.2, respectively). PS levels of up to 13.0 mg kg−1 (mean 4.3 mg kg−1) were exclusively found in mulched topsoil. Similarly to PP, the high variation in PS levels across sites rendered statistical differences insignificant (Student’s t = −1.000, p = 0.374, df = 4, Cohen’s d = −0.8). Mulched subsoil and control sites did not show any PS.

Figure 5 Mean PE, PP, and PS microplastic contents (≤ 2 mm) in fields with and without PE mulching; range bars indicate minimum and maximum values of the three sites studies per treatment.

Discussion

So far, only few studies have reported macroplastics counts on arable land. Piehl et al. (2018), for instance, identified 206 macroplastic fragments ha−1 on non-mulched soils in southern Germany. 34 years of sewage sludge applications resulted in 637 items ha−1 (Weber, Santowski & Chifflard, 2022). By contrast, a recent study by Li et al. (2022) found a tremendous number of 0.65 million items ha−1 in Chinese fields covered with plastic mulches for the previous 32 years. With 89–206 items ha−1, our macroplastic counts were less than or equal to those reported by Piehl et al. (2018) and thereby considerably lower than the macroplastic emissions from Chinese long-term plastic mulching (Li et al., 2022). This is interesting because our study sites were still mulched with plastic films for the last three years and yet showed comparatively low macroplastic counts. Our non-mulched controls hardly showed any macroplastic contamination (0–75 items ha−1). These differences may be attributed to different film thicknesses used in Germany and China. While EU regulations require a minimum mulch film thickness of 20 µm (EN 13655, 2018), Chinese mulches may be as thin as 6–8 µm (Zhang et al., 2016). Such thin films are typically less durable and may favor the formation of plastic debris. Moreover, an increasing public awareness of plastic pollution in recent years (Heidbreder et al., 2019) may have encouraged farmers to collect visible plastic fragments from their fields, which potentially reduced our macroplastic findings further.

With 2.3 particles kg−1, the number of mesoplastic particles we identified in mulched topsoil was about three times higher than in non-mulched controls. Comparable mesoplastic numbers in topsoil and subsoil indicated a certain degree of particle translocation towards deeper soil, for example by plowing. The most abundant polymers were PE and PP, which reflects the polymeric composition of the used PE and PP covers. In line with our results, Harms et al. (2021) observed mean particle counts of 4 ± 11 items kg−1 in the plowing horizon (0–30 cm) of fields in northern Germany that were partially mulched with plastic films or amended with sewage sludge. German cropland without any previous plastic mulching or sewage sludge amendments only contained 0.3 ± 0.4 mesoplastic particles kg−1 (Piehl et al., 2018). In China, however, Liu et al. (2018) identified 3–7 mesoplastics kg−1 in mulched topsoil (0–6 cm), while Zhang & Liu (2018) reported up to 1,600 particles kg−1 in the uppermost 10 cm of the soil. Both studies were conducted in areas known for intensive mulching practices in the previous 10–25 years. In this respect, and in line with our macroplastic findings, the duration and intensity of plastic mulch applications seem to be decisive for the extent of soil pollution with macroplastics and mesoplastics and the potential for long-term plastic accumulation. A similar relationship has already been suggested for microplastics in plastic-mulched soil (Huang et al., 2020).

The levels of PE, PP, and PS microplastics we detected in fields with mulch film management were considerably higher than in the respective controls. This trend was particularly pronounced in the topsoil. The subsoil hardly contained any PE or PS. However, PP contents in topsoil and subsoil almost leveled which corresponds to our mesoplastic findings. This may indicate the selective translocation of PP particles towards deeper soil. However, even our maximum values of 0.55 mg kg−1 PE and 1.2 mg kg−1 PP were close to our method LOD, which makes it hard to draw definite conclusions. Similar to our microplastic results, a meta analysis by Büks & Kaupenjohann (2020) reported microplastic contents in plastic-mulched soil ranging from 0.1–1.2 mg kg−1. Sewage sludge applications led to microplastic contents of 1.4–5.8 mg kg−1(Büks & Kaupenjohann, 2020). However, plastic masses were converted from particle sizes and densities, which is increasingly discouraged for its high estimate errors (Thomas et al., 2020). One of the few mass-based quantifications found 915 ± 63 mg kg−1 PE, PP, and PS in roadside soil (Dierkes et al., 2019). By contrast, Steinmetz et al. (2022) only reported 3–35 mg kg−1 PE and 5–10 mg kg−1 PP in plastic-mulched topsoil. The highest values were found in fields covered with 40 µm perforated foil. Thicker and more durable films hardly emitted any plastics, which is in agreement with our present findings. Although PS is usually not used for agricultural plastic mulching (Steinmetz et al., 2022), we detected up to 13 mg kg−1 PS in previously mulched topsoil. Presumably, these may have originated from additional styrofoam packaging used when transplanting the strawberries and zucchinis grown at the mulched sites as opposed to the non-mulched cornfields. Thus, PS may be considered a viable marker for littering.

All in all, macroplastic, mesoplastic, and microplastic levels in mulched soil were higher than in non-mulched soil. Moreover, macroplastics and mesoplastics shapes and sizes resembled the used plastic films. Comparatively low PE, PP, and PS microplastic contents may be attributed to the rather short-term use of 50 µm thick plastic mulches as opposed to longer-term use of thinner films common in China. For its higher durability, the mulch films applied in our study probably have not led to distinct microplastic formation yet. Nonetheless, our observations corroborate the assumption that plastic mulches fragment into smaller debris. This in line with previous studies that found oxo-degradable PE mulch residues degrading to microplastics when left in soil for 8.5 years (Briassoulis et al., 2015). Similarly, Chinese scholars found soil microplastic counts increasing with 10–25 years of long-term mulching (Huang et al., 2020; Zhang et al., 2020; Feng, Lu & Liu, 2021) but without assessing macroplastics or mesoplastics. Furthermore, we showed that mesoplastics and microplastics are to a certain extent translocated to deeper soil (40 cm). Particle translocation may be caused by plowing, percolation, or bioturbation (Rillig, Ziersch & Hempel, 2017; Li, Song & Cai, 2020; Liu et al., 2018; Zhang et al., 2020). However, sites 1 and 2 planted with strawberries were not plowed for three years which suggest another transport process like bioturbation (Heinze et al., 2021) or a legacy contamination.

Conclusions

The aim of this study was to assess macroplastics, mesoplastics, and microplastics in the topsoil and subsoil of fields previously covered with plastic mulch for three consecutive years. Across all mulched fields, we identified macroplastic film residues that were associated with elevated mesoplastic and microplastic levels. With less than 13 mg kg−1, the detected PE, PP, and PS microplastic contents were low in comparison with previous studies but are likely to increase with future plastic use. Due to the severely limited degradability of conventional polymers like PE and PP, such plastic debris is likely to accumulate in soil over time. Yet, it has not been conclusively addressed how and to what extent plastic debris is translocated into deeper soil. Furthermore, the eco(toxico)logical implications of microplastics at the reported levels still need to be unraveled. Therefore, future long-term studies are required for a more comprehensive understanding of the fate and effects of microplastics in soil.

The authors thank Silvia Eichhöfer, Andreas Hirsch, and Meta Merl for their help in the laboratory and with soil sampling. Gabriele E. Schaumann is kindly acknowledged for her guidance and support.

Additional Information and Declarations

Competing Interests

Author Contributions

Field Study Permissions

Data Availability

The authors declare there are no competing interests.

Zacharias Steinmetz conceived and designed the experiments, performed the experiments, analyzed the data, prepared figures and/or tables, authored or reviewed drafts of the article, and approved the final draft.

Heike Schröder conceived and designed the experiments, performed the experiments, analyzed the data, prepared figures and/or tables, authored or reviewed drafts of the article, and approved the final draft.

The following information was supplied relating to field study approvals (i.e., approving body and any reference numbers):

Permissions for soil sampling were granted by the land owners.

The following information was supplied regarding data availability:

All data and code to reproduce data processing and statistical analysis are available at figshare: Steinmetz, Zacharias; Schröder, Heike (2022): Data from: Plastic debris in plastic-mulched soil—a screening study from western Germany. figshare. Dataset. https://doi.org/10.6084/m9.figshare.19435148.v1.

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
