# Peer review of "Plastic debris in plastic-mulched soil—a screening study from western Germany"

_PeerJ, doi:10.7717/peerj.13781_

## Round 0.1 · original submission · Major Revisions

Three Reviewers have provided their opinion and suggestions for improving the manuscript. Please address the questions and comments in the response letter and make respective changes in the manuscript.

·

Basic reporting

The submitted article is very interesting and suitable for their publication, although I'm not sure about a multidisciplinary journal such as PeerJ. The paper has several new information and potential interest due to the topic's novelty. I like the paper!

However, the paper needs to improve its discussion with other papers because this question is missing in the present version. Ok, the authors compare their data with german fields, but comparison with other countries is scarce. See some examples: https://doi.org/10.5194/soil-6-649-2020 https://doi.org/10.1016/j.scitotenv.2020.141917 or https://doi.org/10.1016/j.scitotenv.2019.03.368 During the reviewing process, i didn't see that this manuscript is a short communication or a technical report. For these reasons, I think that a good discussion is needed. Linked to this question, the authors refer several times to the previous paper, but i think more information can be added here.

L15 (abstract). I suggest to add "Three fields without any visual plastic" We can not talk about without any plastic, considering the multiple plastic sources in the
environment.

L47. "Crop rotations and shorter plastic applications of less than three years appear" Even less time (sometimes 6 monhts). Three years is the expected life-time for plastic greenhouse not for a mulch that usually is applied for each cultivation period (around 3-6 months). I suggest to read (EIP-AGRI Focus Group Reducing the plastic footprint of agriculture) https://ec.europa.eu/eip/agriculture/sites/default/files/eip-agri_fg_plastic_footprint_minipaper_a_final.pdf

L170. "So far, only few studies have investigated macroplastics on arable land." In Germany? I suggest to see https://doi.org/10.5194/soil-6-649-2020 https://doi.org/10.1016/j.scitotenv.2020.141917 or https://doi.org/10.1016/j.scitotenv.2019.03.368 Besides, i suggest to add a comparison table with studies from all countries, and not only Germany. This section needs to be improved—a lot.

Experimental design

L70-79. Please, add species names as in the previous study https://doi.org/10.5194/soil-8-31-2022

L77-79. "None of the six sites has received any compost or sewage sludge applications in recent years. While this explicitly excluded some other potential sources of plastic inputs, littering and wind drift could not be controlled for." I agree with the authors on this question because plastics have been converted into an environmental contaminants such as Pb. However, can you add a bit more information about the study area? Potential contamination from industrial or urban sources?

L80. Only a minor comment: Can you add the study area's location on a Germany map?

L90-94.
- Why only these soil properties and not also pH or other properties? In the previous study, more properties were measured https://doi.org/10.5194/soil-8-31-2022
- "The soil texture on-site was determined in accordance with ASTM D422-63 (2007)." Hydrometer?

L96-100.
- Some researchers are indicating that a sieve tower should be used for soil sieving and plastic separation (5 mm, 2 mm and 1 mm). Why were separated directly by 2mm and not previously by 5 mm?

Why NaCl solution and not other solutions (H2O, CaCl2, ZnCl2, olive oil, etc)? NaCl could be good for LDPE plastics, but in an agricultural field, you could find other plastic items (PVC, PP, PET, etc) that could don't have a result with NaCl. Besides, other authors suggest the use of sequential extractions with different solutions at a different density ranges (e,g, H2O+NaCl, or NaCl+ZnCl2) (e.g. https://doi.org/10.1016/j.scitotenv.2017.10.213 or https://doi.org/10.1016/j.scitotenv.2020.141917) I agree with the authors on the use of NaCl (cheap, environmentally friendly...), but this question should be improved.

L101-112. Usually, organic matter digestion is an issue in agricultural soils, and for these reasons is, used Fenton or H202 extraction to remove organic matter https://doi.org/10.1021/acs.est.8b01517. However, why this approach was not carried out for microplastics? The sodium hexametaphosphate solution can be used as a substitute for Fenton/H2O2 reagent? I'm interested too because I have similar samples.

L107-112. The solutions and reagents were used for which interest? Sample purification? Py-GC/MS analysis? Please, add more information. Although this information could be previously explained (https://doi.org/10.5194/soil-8-31-2022), i suggest improving this information also here. All readers want to read the information in the paper and not read the previous paper ;)

L113. Please, add information about quality controls and measures to avoid plastic contamination in the lab. Steel or glass equipment? A positive control was carried out? This information was indicated in the previous work but not here https://doi.org/10.5194/soil-8-31-2022

L136. No information about statistical analysis.

Validity of the findings

Please, try to compare your data with previous data from other countries (not only Germany).

·

Basic reporting

The manuscript reports a well-defined objective, that is to determine the presence of plastic micro-, meso-, and macroparticles in plastic-mulched soils from agricultural lands. A major shortcoming in the reporting of data is the definitions used for these three types of particles. Throughout the literature there are various ranges used to define macro-, meso-, micro-, and nanoplastics based on the physical dimensions of the particles, which creates a major barrier for comparing results from different studies. A publication by Hartmann et al. in 2019 (Are We Speaking the Same Language? Recommendations for a Definition and Categorization Framework for Plastic Debris https://doi.org/10.1021/acs.est.8b05297) had pointed out the need to use a common terminology in order to address this issue. I believe it is imperative to start using a common framework and suggest that the authors re-group their results using the limits suggested in this publication, i.e. microplastics: 1 - 1,000 µm, mesoplastics: 1 - 10 mm, and macroplastics: larger than 1 cm.
In addition, the concentration of macroplastics is reported as number of particles per area of field, mesoplastics as number of particles per mass of soil, and microplastics as mass of particles per mass of soil. Although it is perfectly understandable that the choice of units derives from the quantification methods used, I suggest that the authors go one step further and make appropriate assumptions in order to provide a qualitative comparison between the three types of particles, using the same units; preferably as number of particles per volume of soil. If the authors feel that other units are more appropriate, please feel free to use them.

Experimental design

The experimental design is very well planned and executed. The choice of reference fields provides the means to take into account the effect of factors other than mulching on the presence of plastics in soil. I would like to see a discussion about the possibilities of sample contamination during handling in order to further strengthen the quality of the experimental work presented.
Line 82: please elaborate on the four-eye principle, either in the main text or as supporting information.
Line 102: please report the concentration and pH of the sodium hexametaphosphate solution used.

Validity of the findings

no comment

Additional comments

This is an excellent field study on a subject with scientific and socioeconomic interest, which I recommend to be published once the issues mentioned above are addressed.

Reviewer 3 ·

Basic reporting

I would like to thank the authors for developing comprehensive but not easy to apply (because of the necessity of a sophisticated instrument) methods for microplastic in soil matrices. The chemical used in the analysis is not also environmentally friendly.
The methodology was already applied in another previously published study by the same authors. However, I have a few concerns about the density separation and the chemicals that they used.

- NaCl based density separation is not suitable for PET or other high-density plastics. They should mention this limitation somewhere in the discussion section.
- Xylene has melting effects on especially polystyrene plastics. How did they tackle this issue? This part needs clarification. Also similar concerns for 1,2,4-trichlorobenzene.
- Please be consistence about units. fragments or particles or pieces or MPs.
- Where do plastic size categories come from? Please be consistent with the literature about size classification.
- As they mentioned that only PE and PP type mulches are used for mulching and then why did they consider PS? This part needs to be clarified.
- The term soil-associated debris is not actually the correct term for soil plastic pollution. If they used this term then we need to talk about non-soil-associated debris as well? Since the authors work on soil plastic pollution then it is not necessary to mention soil-associated debris in the text. Please fix this issue.
- How did they measure the contamination issue? Any blank?
- The plastic analysis part of the paper is not reader-friendly. Please provide a flow chart for explaining each step.

Experimental design

A couple of issues need to be clarified.

1) The corrosive effects issues of chemicals used for py-GC/MS have to be mentioned.
2) The limitation of the NaCl separation method needs to be mentioned?
3) The blank procedure is missing.
4) What about contamination control?

Validity of the findings

Already mentioned in previous section.

---

## Round 0.2 · Minor Revisions

Thank you for revising the manuscript. The Reviewers were mostly satisfied with the revisions made and responses provided but Reviewer 2 had some additional comments. Please address this additional concern before the manuscript can be accepted for publication.

·

Basic reporting

All comments have been addressed. The manuscript quaility has been improved since last submission. New figures and new text in the discussion section improved the manuscript. It can be accepted.

Experimental design

No comments.

Validity of the findings

No comments.

·

Basic reporting

The authors have adequately addressed all comments, except the reporting of the sodium hexametaphosphate solution pH. Even though the pH was not adjusted, the value needs to be reported, because it indicates the mechanism of stabilization of the aqueous suspension, i.e. highly negatively charged particle surfaces inducing electrostatic stabilization. Please report the pH value. No further action is required.

Experimental design

N/A

Validity of the findings

N/A

Additional comments

N/A

---

## Author Rebuttal · Round 0.2

Dear Prof. Dr. Mortimer, dear Reviewers,

Thank you very much for the opportunity to revise. The very constructive reviewer comments helped us to further improve our manuscript. This included the addition of a flow chart to better understand our plastic analysis and a complete rewrite of our discussion section. In our revision, we addressed all reviewer comments and corrected a few minor mistakes. Please find below our detailed responses to the reviewer comments. We hope that the revised manuscript is now acceptable and convinces you and the reviewers.

Kind regards,

Zacharias Steinmetz & Heike Schröder

**Detailed responses to the revision of Steinmetz & Schröder (#2022:03:72342:0:1:REVIEW)**

> Remark: The blue line numbers refer to the numbering in the revised manuscript. Changes to the original text are highlighted in yellow.

**Reviewer: Andrés Rodríguez-Seijo**

*Basic reporting*

The submitted article is very interesting and suitable for their publication, although I'm not sure about a multidisciplinary journal such as PeerJ. The paper has several new information and potential interest due to the topic's novelty. I like the paper!

> Dear Dr. Rodríguez-Seijo, thank you very much for your encouraging feedback. The manuscript was submitted to the special issue 'Nano- and Microplastics in the Environment' (https://peerj.com/special-issues/93-microplastics) and thus screened prior submission to make sure that it fits the scope of the special issue.

However, the paper needs to improve its discussion with other papers because this question is missing in the present version. Ok, the authors compare their data with german fields, but comparison with other countries is scarce. See some

examples: https://doi.org/10.5194/soil-6-649-2020 https://doi.org/10.1016/j.scitotenv.2020.141917 or https://doi.org/10.1016/j.scitotenv.2019.03.368 During the reviewing process, i didn't see that this manuscript is a short communication or a technical report. For these reasons, I think that a good discussion is needed.

We further refined our discussion as suggested. The paragraph on macroplastics now features comparisons with latest findings by Weber et al. (2022, https://doi.org/10.1038/s41598-022-10294-w) and Li et al. (2022, https://doi.org/10.1016/j.envpol.2022.118945). In the microplastics paragraph, we now compare our results with Büks & Kaupenjohann (2020, https://doi.org/10.5194/soil-6-649-2020). Yet, we decided not to include the studies by Corradini et al. (2019, https://doi.org/10.1016/j.scitotenv.2019.03.368; 2021, https://doi.org/10.1016/j.scitotenv.2020.141917) and since they did not focus on plastic mulching and neither reported macroplastics nor microplastics on a mass basis, which makes comparisons with our present study difficult.

Our macroplastic discussion now reads as follows:

> **Lines 187–201:** *So far, only few studies have* ==reported macroplastics counts on arable land.== *Piehl et al. (2018), for instance, identified 206 macroplastic fragments ha–1 on non-mulched soils in southern Germany.* ==34 years of sewage sludge applications resulted in 637 items ha–1 (Weber et al., 2022). By contrast, a recent study by Li et al. (2022) found a tremendous number of 0.65 million items ha–1 in Chinese fields covered with plastic mulches for the previous 32 years. With 89–206 items ha–1 , our macroplastic counts were less than or equal to those reported by Piehl et al. (2018) and thereby considerably lower than the macroplastic emissions from Chinese long-term plastic mulching (Li et al., 2022). This is interesting because our study sites were still mulched with plastic films for the last three years and yet showed comparatively low macroplastic counts.== *Our non-mulched controls hardly showed any macroplastic contamination (0–75 items ha–1 ).* ==These differences may be attributed to different film thicknesses used in Germany and China. While EU regulations require a minimum mulch film thickness of 20 μm (EN 13655, 2018), Chinese mulches may be as thin as 6–8 μm (Zhang et al., 2016). Such thin films are typically less durable and may favor the formation of plastic debris. Moreover, an increasing public awareness of plastic pollution in recent years (Heidbreder et al., 2019) may have encouraged farmers to collect visible plastic fragments from their fields, which potentially reduced our macroplastic findings further.==

> **Lines 222–228:** *Similar to our microplastic results, a meta analysis by Büks and Kaupenjohann (2020) reported microplastic contents in plastic-mulched soil ranging from 0.1–1.2 mg kg–1 . Sewage sludge applications led to microplastic contents of 1.4– 5.8 mg kg–1 (Büks and Kaupenjohann, 2020). However, plastic masses were converted from particle sizes and densities, which is increasingly discouraged for its high estimate errors (Thomas et al., 2020). One of the few mass-based quantifications found 915±63 mg kg–1 PE, PP, and PS in roadside soil (Dierkes et al., 2019).*

Linked to this question, the authors refer several times to the previous paper, but i think more information can be added here.

> In line with your suggestions, we added the following information to the methods section:

> **Lines 100–104:** *Since agricultural plastic covers a primarily made of light-density polymers like PE and PP, mesoplastics were density-separated from non-sieved soil with a saturated NaCl solution (1.2 g cm–3 ). Polymers with a higher density than the applied NaCl solution, such as PET or PVC, were thus systematically excluded from our analysis. To this end, 1 kg of topsoil and subsoil were each mixed with 3 L density solution using a stainless steel whisk.*

> **Lines 107–112:** *Microplastics were extracted in accordance with Steinmetz et al. (2022). In brief, 50 g of sieved soil were dispersed with 125 mL of sodium hexametaphosphate solution (40 g L–1 , CAS 68915-31-1, ≥99% purity, Carl Roth, Karlsruhe, Germany) in a 1 L glass separation funnel and agitated at 150 rpm for 2 h to retrieve plastic debris potentially occluded in soil aggregates. The subsequent addition of 90 g NaCl and 125 mL of distilled water produced a saturated NaCl solution with a density of 1.2 g cm–3 which was agitated for another 2 h.*

> **Lines 114–118:** *The filters were transferred to glass culture tubes (GL18, VWR, Darmstadt Germany), dried at 60 ˚C, and extracted with 8 mL of a 1:1 mixture of p-xylene (CAS 106-42-3, >98.0% purity, Fluka Analytical, München, Germany) and 1,2,4-trichlorobenzene (CAS 120-82-1, 99% purity, Alfa Aesar, Kandel, Germany) at 150 ˚C for 1 h. Polymer dissolution in p-xylene/1,2,4-trichlorobenzene facilitated sample handling and enabled the representative analysis of sample aliquots.*

L15 (abstract). I suggest to add "Three fields without any visual plastic" We can not talk about without any plastic, considering the multiple plastic sources in the environment.

We agree that this statement was problematic with respect to the multitude of external plastic sources. Since the farmers cultivating the fields stated that they haven't applied any plastic products in the past, we clarified the sentence accordingly

> *Line 15: Three fields without any ==reported== plastic use served as control.*

L47. "Crop rotations and shorter plastic applications of less than three years appear" Even less time (sometimes 6 monhts). Three years is the expected life-time for plastic greenhouse not for a mulch that usually is applied for each cultivation period (around 3-6 months). I suggest to read (EIP-AGRI Focus Group Reducing the plastic footprint of agriculture) https://ec.europa.eu/eip/agriculture/sites/default/files/eip-agri_fg_plastic_footprint_minipaper_a_final.pdf

> Thank you for this important remark. The expected lifetime of plastic covers largely depends on the use case and the material properties. In Germany, plastic mulches for strawberry cultivations have a typical film thickness of 50 μm and are used for several growing seasons. We clarified the sentence as follows.

> *Lines 47–48: Crop rotations and shorter plastic applications of ==several months to about three years appear to limit plastic emissions.==*

L170. "So far, only few studies have investigated macroplastics on arable land." In Germany? I suggest to see https://doi.org/10.5194/soil-6-649-2020 https://doi.org/10.1016/j.scitotenv.2020.141917 or https://doi.org/10.1016/j.scitotenv.2019.03.368 Besides, i suggest to add a comparison table with studies from all countries, and not only Germany. This section needs to be improved—a lot.

> We agree that this part required extensive revision. In line with your recommendations, we completely overhauled the first paragraph of the discussion. Since this part focused on macroplastics, we only added references that determined macroplastics in agricultural soil.

> *Lines 187–201: So far, only few studies have ==reported macroplastics counts on arable land.== Piehl et al. (2018), for instance, identified 206 macroplastic fragments ha–1 on non-mulched soils in southern Germany. ==34 years of sewage sludge applications resulted in 637 items ha–1 (Weber et al., 2022). By contrast, a recent study by Li et al. (2022) found a tremendous number of 0.65 million items ha–1 in Chinese fields covered with plastic mulches for the previous 32 years. With 89–206 items ha–1 , our macroplastic counts were less than or equal to those reported by Piehl et al. (2018) and thereby considerably==*

*lower than the macroplastic emissions from Chinese long-term plastic mulching (Li et al., 2022). This is interesting because our study sites were still mulched with plastic films for the last three years and yet showed comparatively low macroplastic counts. Our non-mulched controls hardly showed any macroplastic contamination (0–75 items ha–1 ). These differences may be attributed to different film thicknesses used in Germany and China. While EU regulations require a minimum mulch film thickness of 20 µm (EN 13655, 2018), Chinese mulches may be as thin as 6–8 µm (Zhang et al., 2016). Such thin films are typically less durable and may favor the formation of plastic debris. Moreover, an increasing public awareness of plastic pollution in recent years (Heidbreder et al., 2019) may have encouraged farmers to collect visible plastic fragments from their fields, which potentially reduced our macroplastic findings further.*

However, we are reluctant to add a comparison table since they are more typical for review papers.

*Experimental design*

L70-79. Please, add species names as in the previous study https://doi.org/10.5194/soil-8-31-2022

We added the Latin species names as suggested.

***Lines 73–79:*** *Sites 1 and 2 were cultivated with strawberries (Fragaria x ananassa) continuously covered with black PE mulch (50 µm film thickness) for the previous three years. Site 3 was planted with zucchinis (Cucurbita pepo) and mulched with black PP of the same film thickness during the last three growing seasons. Sites 4–6 were situated near Pulheim (51° 0' N 6° 47' E), Stommelerbusch (51° 4' N 6° 46' E), and Rommerskirchen (51° 2' N 6° 42' E), and served as reference sites without any previous plastic use. Sites 4 and 6 were cornfields (Triticum aestivum); site 5 was cultivated with potatoes (Solanum tuberosum).*

L77-79. "None of the six sites has received any compost or sewage sludge applications in recent years. While this explicitly excluded some other potential sources of plastic inputs, littering and wind drift could not be controlled for." I agree with the authors on this question because plastics have been converted into an environmental contaminants such as Pb. However, can you add a bit more information about the study area? Potential contamination from industrial or urban sources?

We added this information as suggested.

> *Lines 80–81:* ==*The closest urban or industrial areas were at least 5 km away from the experimental sites.*==

L80. Only a minor comment: Can you add the study area's location on a Germany map?

> We added a small Germany map to better show the map section; see **Figure 1** for details.

L90-94.

- Why only these soil properties and not also pH or other properties? In the previous study, more properties were measured https://doi.org/10.5194/soil-8-31-2022

> We also measured pH, but with handheld probes only. We did not find it informative enough to add it this time. **Table 1** now also features the pH values in the fields. We amended the text accordingly.

> > *Lines 97–98:* ==*The soil pH was measured in the field using a handheld probe (X4 Life 4in1, Braunschweig, Germany).*==

> > *Lines 158–159:* ==*The soil pH was circumneutral ranging from 6–7 across all sites.*==

- "The soil texture on-site was determined in accordance with ASTM D422-63 (2007)." Hydrometer?

> Yes. We added this information.

> > *Lines 95–96:* *The soil texture on site was determined* ==*using a soil hydrometer*== *in accordance with ASTM D422-63 (2007).*

L96-100.

- Some researchers are indicating that a sieve tower should be used for soil sieving and plastic separation (5 mm, 2 mm and 1 mm). Why were separated directly by 2mm and not previously by 5 mm?

> Soil analyses are typically performed on fine soil ≤2 mm and analyte contents are based on fine soil as common reference. We agree that this contrasts common microplastic definitions. However, interpreting the fate of 5 mm plastic particles in a 2 mm sample matrix seems like a paradox. Moreover, distinguishing between 1 and 2 mm particles in a heterogeneous, particulate matrix may be challenging in terms of its environmental relevance. Therefore, we decided to use a common basis both for fine soil and

microplastic analyses. To make this comprehensible to the reader, we added the following clarification to the introduction:

> **Lines 57–59**: Notwithstanding common size classifications of plastic debris (Hartmann et al., 2019), we chose 2 mm instead of 1 mm as the upper size limit for microplastics to be consistent with soil analyses usually performed on fine soil ≤2 mm (Thomas et al., 2020).

In agreement with comments by the other two reviewers, however, we decided to restructure our mesoplastics and macroplastics data to fit common size classifications of 1 cm for macroplastics. This has been changed throughout the manuscript, and includes adjustments of **Figures 3 and 4** as well as in the results section:

> **Lines 161–172**: A total of 35 macroplastic items were collected from surface soil while following the predefined sampling transects. With respect to the sampled area, this was equivalent to 89–206 fragments ha–1 at sites 1–3, which were previously covered with mulch films. 75 fragments ha–1 were found at the control site 5. Sites 4 and 6 did not show any macroplastic contamination. 80% of the collected items were identified as black PE foils. In addition to that, we found rope fragments, clips, and residues of blue and white plastic films. The total number of mesoplastic particles extracted from non-sieved soil was 17. Plastic-mulched topsoil and subsoil each contained 2.3 plastic particles kg–1 (Figure 3). With 1.3 particles kg–1 , the majority of particles were films, followed by fragments (0.7 particles kg–1 ) and fibers (0.3 particles kg–1 ) of blue and black color (items 1–3, 6, 7, 12–14, 15, and 17, Figure 4). The predominant polymers were identified as PP, PE, and PS (r >0.83, Figure 3). In non-mulched controls, only 1.0 particles kg–1 were found in the topsoil. These were a resin-like bead, a black PE film, and a PE fiber (r >0.83, Figure 3; items 8–9, Figure 4). The non-mulched subsoil was free of mesoplastics.

Why NaCl solution and not other solutions ($H_2O$, $CaCl_2$, $ZnCl_2$, olive oil, etc)? NaCl could be good for LDPE plastics, but in an agricultural field, you could find other plastic items (PVC, PP, PET, etc) that could don't have a result with NaCl. Besides, other authors suggest the use of sequential extractions with different solutions at a different density ranges (e,g, $H_2O$+NaCl, or NaCl+$ZnCl_2$) (e.g. https://doi.org/10.1016/j.scitotenv.2017.10.213 or https://doi.org/10.1016/j.scitotenv.2020.141917) I agree with the authors on the use of NaCl (cheap, environmentally friendly...), but this question should be improved.

The focus of our study was to quantify the extent to which agricultural plastic covers made of PE and PP function as a source for plastic debris in soil. Since both are light-density polymers, we decided to resort to a more ecofriendly density solution. We now acknowledge this in the methods section.

> **Lines 100–103:** *Since agricultural plastic covers a primarily made of light-density polymers like PE and PP, mesoplastics were density-separated from non-sieved soil with a saturated NaCl solution (1.2 g cm–3 ). Polymers with a higher density than the applied NaCl solution, such as PET or PVC, were thus systematically excluded from our analysis.*

L101-112. Usually, organic matter digestion is an issue in agricultural soils, and for these reasons is, used Fenton or H2O2 extraction to remove organic matter https://doi.org/10.1021/acs.est.8b01517. However, why this approach was not carried out for microplastics? The sodium hexametaphosphate solution can be used as a substitute for Fenton/H2O2 reagent? I'm interested too because I have similar samples.

The sodium hexametaphosphate solution served the purpose of dispersing soil aggregates which enables the extraction of plastic debris occluded in soil aggregates. We added this information:

> **Lines 107–110:** *In brief, 50 g of sieved soil were dispersed with 125 mL of sodium hexametaphosphate solution (40 g L–1 , CAS 68915-31-1, ≥99% purity, Carl Roth, Karlsruhe, Germany) in a 1 L glass separation funnel and agitated at 150 rpm for 2 h to retrieve plastic debris potentially occluded in soil aggregates.*

Our solvent-based Py–GC/MS approach was shown to be robust against soil organic carbon content <2.5% as detailed in our previous publication. An additional oxidative sample preparation step was thus not necessary.

L107-112. The solutions and reagents were used for which interest? Sample purification? Py-GC/MS analysis? Please, add more information. Although this information could be previously explained (https://doi.org/10.5194/soil-8-31-2022), i suggest improving this information also here. All readers want to read the information in the paper and not read the previous paper ;)

The organic solvent mixture was used to dissolve the polymers prior Py–GC/MS analysis. This facilitated sample handling, reduced the risk of contamination, and allowed for the simple and representative aliquotation of sample extracts. We agree that this information is useful to the readers and added it as suggested.

> *Lines 117–121:* *Polymer dissolution in p-xylene/1,2,4-trichlorobenzene facilitated sample handling and enabled the representative analysis of sample aliquots.* *After cooling down, the extracts were thus transferred to glass vials with polytetrafluoroethylene-sealed caps for subsequent quantification of PE, PP, and PS via Py–GC/MS. The recovery of the extraction procedure was 86–105% (Steinmetz et al., 2022).*

L113. Please, add information about quality controls and measures to avoid plastic contamination in the lab. Steel or glass equipment? A positive control was carried out? This information was indicated in the previous work but not here https://doi.org/10.5194/soil-8-31-2022

> We added a QA section.

> *Lines 146–151:* *To reduce the risk of contamination, all sampling and laboratory equipment coming into direct contact with the samples or extract solutions was made of glass, stainless steel, or PTFE. The worn laboratory coats were of 100% cotton. All samples and extracts were kept in closed vessels or covered with aluminum foil or watch glasses whenever possible.*

> *In addition, a potential plastic contamination during sample handling was monitored with procedural blanks, which went through all sample preparation steps but without soil addition.*

L136. No information about statistical analysis.

> Our statistical analysis is included the section named 'plastic analysis'. We added a few more details.

> *Lines 142–144:* *Microplastic contents between plastic-mulched fields and non-mulched controls were normally distributed and homogeneous in variance and thus statistically analyzed using Student's t tests. Effect sizes were measured with Cohen's d* *using R statistical software (version 4.2.0)..*

*Validity of the findings*

Please, try to compare your data with previous data from other countries (not only Germany).

> As highlighted above, we now discuss our results with more studies, including some from other countries. For instance

> *Lines 188–191:* *34 years of sewage sludge applications resulted in 637 items ha–1 (Weber et al., 2022). By contrast, a recent study by Li et al. (2022) found a tremendous*

*number of 0.65 million items ha–1 in Chinese fields covered with plastic mulches for the previous 32 years.*

***Lines 196–198:** These differences may be attributed to different film thicknesses used in Germany and China. While EU regulations require a minimum mulch film thickness of 20 µm (EN 13655, 2018), Chinese mulches may be as thin as 6–8 µm (Zhang et al., 2016).*

***Lines 222–228:** Similar to our microplastic results, a meta analysis by Büks and Kaupenjohann (2020) reported microplastic contents in plastic-mulched soil ranging from 0.1–1.2 mg kg–1 . Sewage sludge applications led to microplastic contents of 1.4–5.8 mg kg–1 (Büks and Kaupenjohann, 2020). However, plastic masses were converted from particle sizes and densities, which is increasingly discouraged for its high estimate errors (Thomas et al., 2020). One of the few mass-based quantifications found 915±63 mg kg–1 PE, PP, and PS in roadside soil (Dierkes et al., 2019).*

**Reviewer: Andreas Gondikas**

*Basic reporting*

The manuscript reports a well-defined objective, that is to determine the presence of plastic micro-, meso-, and macroparticles in plastic-mulched soils from agricultural lands.

Dear Dr. Gondikas, thank you very much for this valuable feedback.

A major shortcoming in the reporting of data is the definitions used for these three types of particles. Throughout the literature there are various ranges used to define macro-, meso-, micro-, and nanoplastics based on the physical dimensions of the particles, which creates a major barrier for comparing results from different studies. A publication by Hartmann et al. in 2019 (Are We Speaking the Same Language? Recommendations for a Definition and Categorization Framework for Plastic Debris https://doi.org/10.1021/acs.est.8b05297) had pointed out the need to use a common terminology in order to address this issue. I believe it is imperative to start using a common framework and suggest that the authors re-group their results using the limits suggested in this publication, i.e. microplastics: 1 - 1,000 µm, mesoplastics: 1 - 10 mm, and macroplastics: larger than 1 cm.

We agree that our microplastic size range (≤2 mm) contrasts common microplastic definitions of 1–1000 µm or 1–5000 µm. However, soil analyses are typically performed on fine soil ≤2 mm and analyte contents are based on fine soil as common reference.

Interpreting the fate of 5 mm particles in a 2 mm sample matrix seems like a paradox. Moreover, distinguishing between 1 and 2 mm particles in a heterogeneous, particulate matrix may be challenging in terms of its environmental relevance. Therefore, we decided to use a common basis both for fine soil and microplastic analyses. We added the following statement to the introduction, to make our rationale comprehensible to the reader.

> **Lines 57–59:** *Notwithstanding common size classifications of plastic debris (Hartmann et al., 2019), we chose 2 mm instead of 1 mm as the upper size limit for microplastics to be consistent with soil analyses usually performed on fine soil ≤2 mm (Thomas et al., 2020).*

For mesoplastics and macroplastics, we were able to regroup our data as suggested. We now analyze mesoplastics between 2 mm and 1 cm and macroplastics >1 cm and changed the manuscript accordingly. This includes **Figures 3 and 4** as well as the results section. The regrouping did not affect the interpretation of our results so that the discussion remained unchanged.

> **Lines 161–172:** *A total of 35 macroplastic items were collected from surface soil while following the predefined sampling transects. With respect to the sampled area, this was equivalent to 89–206 fragments ha–1 at sites 1–3, which were previously covered with mulch films. 75 fragments ha–1 were found at the control site 5. Sites 4 and 6 did not show any macroplastic contamination. 80% of the collected items were identified as black PE foils. In addition to that, we found rope fragments, clips, and residues of blue and white plastic films. The total number of mesoplastic particles extracted from non-sieved soil was 17. Plastic-mulched topsoil and subsoil each contained 2.3 plastic particles kg–1 (Figure 3). With 1.3 particles kg–1 , the majority of particles were films, followed by fragments (0.7 particles kg–1 ) and fibers (0.3 particles kg–1 ) of blue and black color (items 1–3, 6, 7, 12–14, 15, and 17, Figure 4). The predominant polymers were identified as PP, PE, and PS (r >0.83, Figure 3). In non-mulched controls, only 1.0 particles kg–1 were found in the topsoil. These were a resin-like bead, a black PE film, and a PE fiber (r >0.83, Figure 3; items 8–9, Figure 4). The non-mulched subsoil was free of mesoplastics.*

In addition, the concentration of macroplastics is reported as number of particles per area of field, mesoplastics as number of particles per mass of soil, and microplastics as mass of particles per mass of soil. Although it is perfectly understandable that the choice of units

derives from the quantification methods used, I suggest that the authors go one step further and make appropriate assumptions in order to provide a qualitative comparison between the three types of particles, using the same units; preferably as number of particles per volume of soil. If the authors feel that other units are more appropriate, please feel free to use them.

We generally agree that it is important to keep units comparable to each other. In our case though, this would involve high uncertainties. Firstly, reporting analyte concentrations on a volume basis is not common in soil science since the soil's bulk density is highly variable. Secondly, macroplastics collected from the soil surface usually stay on the soil surface so that referring to a volume or mass of soil would be highly error-prone. For macro- and mesoplastics, we further aimed to stick to commonly used units to facilitate comparisons with other studies.

Yet, we too would have been eager to relate macroplastics to meso- and microplastic levels. Even with inconsistent units, such relations are still possible using correlation matrices. However, the limited number of our experimental sites hampers such statistical analyses as exemplarily shown below for our data:

[Figure]

Therefore, we would suggest to keep comparisons between macroplastics, mesoplastics, and microplastics rather on a qualitative level.

*Experimental design*

The experimental design is very well planned and executed. The choice of reference fields provides the means to take into account the effect of factors other than mulching on the presence of plastics in soil. I would like to see a discussion about the possibilities of sample contamination during handling in order to further strengthen the quality of the experimental work presented.

Thank you for this important remark. We added a QA section as suggested.

**Lines 146–151:** *To reduce the risk of contamination, all sampling and laboratory equipment coming into direct contact with the samples or extract solutions was made of glass, stainless steel, or PTFE. The worn laboratory coats were of 100% cotton. All samples and extracts were kept in closed vessels or covered with aluminum foil or watch glasses whenever possible.*

*In addition, a potential plastic contamination during sample handling was monitored with procedural blanks, which went through all sample preparation steps but without soil addition.*

Line 82: please elaborate on the four-eye principle, either in the main text or as supporting information.

We added some additional information together with a reference explaining the principle.

**Lines 85–87:** *Macroplastic fragments >1 cm were collected from surface soil at a maximum distance of 1 m from either side of the diagonals, based on the four-eye principle to decrease the risk of overlooking plastic debris (Prume et al., 2021).*

Line 102: please report the concentration and pH of the sodium hexametaphosphate solution used.

We added the concentration of the sodium hexametaphosphate solution as suggested. The pH was not adjusted at left as is.

**Lines 107–110:** *In brief, 50 g of sieved soil were dispersed with 125 mL of sodium hexametaphosphate solution (40 g $L^{-1}$ , CAS 68915-31-1, ≥99% purity, Carl Roth,*

*Karlsruhe, Germany)* *in a 1 L glass separation funnel and agitated at 150 rpm for 2 h to retrieve plastic debris potentially occluded in soil aggregates..*

*Additional comments*

This is an excellent field study on a subject with scientific and socioeconomic interest, which I recommend to be published once the issues mentioned above are addressed.

**Reviewer #3**

*Basic reporting*

I would like to thank the authors for developing comprehensive but not easy to apply (because of the necessity of a sophisticated instrument) methods for microplastic in soil matrices. The chemical used in the analysis is not also environmentally friendly.

> Thank you for your positive feedback. We agree on the environmental friendliness of our solvents and continuously thrive for reducing the required amounts and replacing it with viable, more ecofriendly alternatives.

The methodology was already applied in another previously published study by the same authors. However, I have a few concerns about the density separation and the chemicals that they used.

- NaCl based density separation is not suitable for PET or other high-density plastics. They should mention this limitation somewhere in the discussion section.

> The focus of our study was to quantify the extent to which agricultural plastic covers made of PE and PP function as a source for plastic debris in soil. Since both are light-density polymers, we decided to resort to a more ecofriendly density solution. We now acknowledge this in the methods section.

> > **Lines 100–103:** *Since agricultural plastic covers a primarily made of light-density polymers like PE and PP, mesoplastics were density-separated from non-sieved soil with a saturated NaCl solution (1.2 g cm–3 ). Polymers with a higher density than the applied NaCl solution, such as PET or PVC, were thus systematically excluded from our analysis.*

- Xylene has melting effects on especially polystyrene plastics. How did they tackle this issue? This part needs clarification. Also similar concerns for 1,2,4-trichlorobenzene.

The organic solvent mixture was used to dissolve the polymers prior Py–GC/MS analysis. This facilitated sample handling, reduced the risk of contamination, and allowed for the simple and representative aliquotation of sample extracts. We agree that this information is useful to the readers and added it as suggested.

> **Lines 117–121:** *Polymer dissolution in p-xylene/1,2,4-trichlorobenzene facilitated sample handling and enabled the representative analysis of sample aliquots. After cooling down, the extracts were thus transferred to glass vials with polytetrafluoroethylene-sealed caps for subsequent quantification of PE, PP, and PS via Py–GC/MS. The recovery of the extraction procedure was 86–105% (Steinmetz et al., 2022).*

- Please be consistence about units. fragments or particles or pieces or Mps.

We generally agree that it is important to keep units comparable to each other, we suggest to keep them in this particular case. For our macro- and mesoplastics, we aimed at sticking to commonly used units to facilitate comparisons with other studies. These are particles per kg for mesoplastics and particles per hectare for macroplastics. On the contrary, microplastics <1 mm are still very difficult to assess on a particle basis. For this reason, we resorted to mass-based methods. Accordingly, we reported the results in mg/kg. Please also note that particle-to-mass conversions are increasingly discouraged in the scientific literature for its high susceptibility to errors (for instance Thomas et al. 2020, https://doi.org/10.3390/su12219074 or Braun et al. 2018, https://bmbf-plastik.de/en/publication/discussion-paper-microplastics-analytics).

- Where do plastic size categories come from? Please be consistent with the literature about size classification.

We regrouped our mesoplastics and macroplastics data to fit common size classifications and now analyze mesoplastics between 2 mm and 1 cm and macroplastics >1 cm and changed the manuscript accordingly. This includes **Figures 3 and 4** as well as the results section. The regrouping did not affect the interpretation of our results so that the discussion remained unchanged.

> **Lines 161–172:** *A total of 35 macroplastic items were collected from surface soil while following the predefined sampling transects. With respect to the sampled area, this was equivalent to 89–206 fragments ha–1 at sites 1–3, which were previously covered with mulch films. 75 fragments ha–1 were found at the control site 5. Sites 4 and 6 did not show any macroplastic contamination. 80% of the collected items were identified as*

*black PE foils. In addition to that, we found rope fragments, clips, and residues of blue and white plastic films. The total number of mesoplastic particles extracted from non-sieved soil was* ==17==. *Plastic-mulched topsoil and subsoil* ==each contained 2.3 plastic particles kg–1 (Figure 3)==. *With 1.3 particles kg–1 , the majority of particles were films, followed by fragments (0.7 particles kg–1 ) and fibers (0.3 particles kg–1 ) of blue and black color* ==(items 1–3, 6, 7, 12–14, 15, and 17, Figure 4)==. *The predominant polymers were identified as PP, PE, and PS (r >0.83, Figure 3). In non-mulched controls, only 1.0 particles kg–1 were found in the topsoil. These were a resin-like bead, a black PE film, and a PE fiber (r >0.83, Figure 3; items 8–9, Figure 4). The non-mulched subsoil was free of mesoplastics.*

However, soil analyses are typically performed on fine soil ≤2 mm and analyte contents are based on fine soil as common reference.  Therefore, we decided to use a common basis both for fine soil and microplastic analyses (2 mm). This is now explained in our introduction.

*__Lines 57–59__: ==Notwithstanding common size classifications of plastic debris (Hartmann et al., 2019), we chose 2 mm instead of 1 mm as the upper size limit for microplastics to be consistent with soil analyses usually performed on fine soil ≤2 mm (Thomas et al., 2020).==*

- As they mentioned that only PE and PP type mulches are used for mulching and then why did they consider PS? This part needs to be clarified.

We added PS as a marker for littering. This is now detailed in the manuscript.

*__Lines 234–235__: ==Thus, PS may be considered a viable marker for littering..==*

- The term soil-associated debris is not actually the correct term for soil plastic pollution. If they used this term then we need to talk about non-soil-associated debris as well? Since the authors work on soil plastic pollution then it is not necessary to mention soil-associated debris in the text. Please fix this issue.

We agree. The term was omitted throughout the manuscript.

- How did they measure the contamination issue? Any blank?

We added a QA section as suggested.

*__Lines 146–151__: ==To reduce the risk of contamination, all sampling and laboratory equipment coming into direct contact with the samples or extract solutions was made of==*

*glass, stainless steel, or PTFE. The worn laboratory coats were of 100% cotton. All samples and extracts were kept in closed vessels or covered with aluminum foil or watch glasses whenever possible.*

*In addition, a potential plastic contamination during sample handling was monitored with procedural blanks, which went through all sample preparation steps but without soil addition.*

- The plastic analysis part of the paper is not reader-friendly. Please provide a flow chart for explaining each step.

Thank you for this important remark. We now added a flow chart (**Figure 2**) to make the methods more comprehensible.

*Experimental design*

A couple of issues need to be clarified.

1) The corrosive effects issues of chemicals used for py-GC/MS have to be mentioned.

As detailed above, the chemicals were used for polymer dissolution. This is now clarified.

**Lines 117–121:** *Polymer dissolution in p-xylene/1,2,4-trichlorobenzene facilitated sample handling and enabled the representative analysis of sample aliquots. After cooling down, the extracts were thus transferred to glass vials with polytetrafluoroethylene-sealed caps for subsequent quantification of PE, PP, and PS via Py–GC/MS. The recovery of the extraction procedure was 86–105% (Steinmetz et al., 2022).*

2) The limitation of the NaCl separation method needs to be mentioned?

The focus of our study was to quantify light-density PE and PP from agricultural plastic covers. This is now clarified as follows.

**Lines 100–103:** *Since agricultural plastic covers a primarily made of light-density polymers like PE and PP, mesoplastics were density-separated from non-sieved soil with a saturated NaCl solution (1.2 g cm–3 ). Polymers with a higher density than the applied NaCl solution, such as PET or PVC, were thus systematically excluded from our analysis.*

3) The blank procedure is missing.

This is now stated in the newly added QA section (see above).

4) What about contamination control?

This is now stated in the newly added QA section (see above).

---

## Round 0.3 · accepted · Accept

Thank you for carefully addressing all the Reviewers' and PeerJ editorial concerns. The manuscript is now suitable for publication.